# Measuring Sleep Quality in the Hospital Environment with Wearable and Non-Wearable Devices in Adults with Stroke Undergoing Inpatient Rehabilitation

**DOI:** 10.3390/ijerph20053984

**Published:** 2023-02-23

**Authors:** Michael Pellegrini, Natasha A. Lannin, Richelle Mychasiuk, Marnie Graco, Sharon Flora Kramer, Melita J. Giummarra

**Affiliations:** 1Department of Neuroscience, The Alfred Centre, Monash University, Melbourne, VIC 3004, Australia; 2Alfred Health, Melbourne, VIC 3053, Australia; 3Institute for Breathing and Sleep, Austin Health, Melbourne, VIC 3084, Australia; 4Department of Physiotherapy, The University of Melbourne, Parkville, VIC 3010, Australia; 5Institute for Health Transformation, Deakin University, Melbourne, VIC 3125, Australia

**Keywords:** stroke, fatigue, activity threshold, awakenings, level of agreement, rehabilitation

## Abstract

Sleep disturbances are common after stroke and may affect recovery and rehabilitation outcomes. Sleep monitoring in the hospital environment is not routine practice yet may offer insight into how the hospital environment influences post-stroke sleep quality while also enabling us to investigate the relationships between sleep quality and neuroplasticity, physical activity, fatigue levels, and recovery of functional independence while undergoing rehabilitation. Commonly used sleep monitoring devices can be expensive, which limits their use in clinical settings. Therefore, there is a need for low-cost methods to monitor sleep quality in hospital settings. This study compared a commonly used actigraphy sleep monitoring device with a low-cost commercial device. Eighteen adults with stroke wore the Philips Actiwatch to monitor sleep latency, sleep time, number of awakenings, time spent awake, and sleep efficiency. A sub-sample (n = 6) slept with the Withings Sleep Analyzer in situ, recording the same sleep parameters. Intraclass correlation coefficients and Bland–Altman plots indicated poor agreement between the devices. Usability issues and inconsistencies were reported between the objectively measured sleep parameters recorded by the Withings device compared with the Philips Actiwatch. While these findings suggest that low-cost devices are not suitable for use in a hospital environment, further investigations in larger cohorts of adults with stroke are needed to examine the utility and accuracy of off-the-shelf low-cost devices to monitor sleep quality in the hospital environment.

## 1. Introduction

Reports of sleep disturbances are common in adults with stroke and are particularly evident during the acute recovery phase while in hospital [1,2]. Sleep problems are experienced by up to two-thirds of people after having a stroke [3], with 25 to 85% of people experiencing persistent fatigue [4]. Importantly, poor sleep quality has been found to affect neuroplasticity and memory consolidation post-stroke [5,6], which may have an adverse causal impact on a range of recovery outcomes. Moreover, poor sleep is associated with poorer motor function [7] and lower levels of physical activity as well as higher levels of fatigue during inpatient rehabilitation [8]. Finally, higher levels of sleep disruption are associated with slower recovery of functional independence and motor recovery throughout inpatient rehabilitation [1,9]. Sleep disruption is common during a hospital stay, regardless of the reason for admission, most often due to clinical care interventions and the noisy environment [10]. However, these impacts may have a particularly damaging effect for people who have had a stroke by impeding neurological recovery while also reducing the level of energy needed to enable optimal participation in rehabilitation [11].

Despite the known impact of sleep quality on stroke outcomes, sleep monitoring and interventions in the hospital environment during inpatient rehabilitation are not routine. In order to improve clinical outcomes, it is important that we have access to simple and valid methods for sleep monitoring. Wearable actigraphy devices are non-invasive technologies that can monitor and accurately measure objective sleep parameters. These devices have been widely incorporated for the measurement of human biometrics, as they can be used to monitor sleep and activity levels in a range of settings without causing discomfort to the wearer, including the assessment of sleep following stroke [12]. However, these monitors are relatively expensive (~USD 2000 each), preventing their routine use in research and clinical settings [13]. Commercial sleep monitoring devices are becoming more readily available on the personalized-device market and may offer low-cost alternatives to monitor sleep quality [14,15]. Consumer devices for measuring physiological and behavioral signals (e.g., heart rate, respiration, and bodily movements) in order to estimate sleep are either worn (e.g., on the wrist), or they are placed under the mattress or in the same room near the bed. In the context of neurological rehabilitation, devices placed under or near the person whose sleep is being monitored may enable superior assessment of sleep, particularly if they have impairments that affect the movement of specific limbs. The Withings Sleep Analyzer (WSA) is one such “nearable” device (~USD 200 each) that is placed under the mattress to detect body movements, cardiac activity, breathing patterns, and snoring. However, while the WSA records several aspects of sleep (e.g., sleep onset and number and duration of wakenings after sleep onset), it has been used primarily to measure decreases in or cessation of breathing during sleep for people with suspected obstructive sleep apnea syndrome [16], and it has not yet been validated in other clinical populations or settings for the measurement of sleep quality or quantity. Before the device is to be used more widely, it is important to investigate its accuracy in monitoring sleep relative to more common validated actigraphy devices. Therefore, this study aimed to describe device usability and to explore the level of agreement between a validated actigraphy device and the WSA within a sample of adults undergoing hospital-based rehabilitation after having a stroke. 

## 2. Materials and Methods

### 2.1. Study Design

The design was a cross-sectional within-subject cohort study. Ethics was approved by the Alfred Health Human Research Ethics Committee (Project ID: 660/21). 

### 2.2. Study Participants

Adults with stroke undergoing inpatient rehabilitation were recruited from two wards: a general medicine rehabilitation ward and a specialized neurological rehabilitation ward. Participants were recruited from two separate wards to capture the differing environments in which adults typically undergo rehabilitation after having a stroke. Adults undergoing stroke rehabilitation on the specialized neurological ward slept in private rooms with more controlled lighting, whereby lighting could be independently dimmed to facilitate onset of sleep. Conversely, adults undergoing rehabilitation on general rehabilitation wards shared a room with one other patient and did not have access to controlled lighting. The potential for noise disturbance was also higher in this shared ward environment. Participants were eligible for the study if they had a diagnosis of stroke, were able to move and roll in bed independently, and did not have any cognitive deficits impacting their capacity to understand and use the sleep monitoring devices. 

### 2.3. Procedure

Sleep was monitored for one night. Participants wore the Philips Actiwatch Spectrum 2 (Philips Respironics, Pittsburgh, PA, USA) on their wrist, while the WSA (Withings, Paris, France) was placed under their mattress. The Actiwatch and WSA devices were retrieved by the researchers the following morning to extract sleep data. Participants were also asked to document the time they got into bed with the intention to sleep, the approximate time it took for them to fall asleep, the time they awoke the following morning, and the number of times they awoke overnight. These data were then used to calculate sleep onset latency (SOL), total sleep time (TST), and number of awakenings as described below.

#### 2.3.1. Philips Actiwatch Spectrum 2

The Actiwatch was placed on the participant’s wrist on the hemiparetic side. Placing the device on the hemiplegic side ensured that participants could safely and independently remove the device with their unaffected limb if necessary. The Actiwatch software (Actiware version 6.0, Philips Respironics, OR, USA) automatically determined sleep onset and offset times via pre-determined activity thresholds [17]. Within the lights off/lights on times, sleep onset time was classified as the first minute of a 10-minute immobile period with <2 activity counts in any 30-s period [18,19]. Ten consecutive minutes of activity was defined as sleep offset [18,19]. During sleep, activity threshold counts >40 per 30-s epoch were defined as awake. This allowed calculation of the number of awakenings and amount of awake time [18,19]. 

#### 2.3.2. Withings Sleep Analyzer

The WSA is an air-inflated sensor mat that detects body and chest movements and respiration vibrations [16]. Once placed under the mattress, the WSA was paired to application-based software (Version 2151, Health mate, Withings, Paris, France) for data collection and storage. The WSA epoch times and activity thresholds used to detect and calculate sleep and wake are not readily available; however, the mat reportedly detects body movements to determine time spent in bed and time registered as awake and asleep [16]. 

### 2.4. Data Collection

#### 2.4.1. Participant Characteristics

Participant characteristics included age, sex, stroke type, stroke location, rehabilitation ward type, days since stroke, and days in rehabilitation at the time of sleep monitoring. No participants were receiving pharmacological support for sleep.

#### 2.4.2. Sleep Parameters and Device Recording

Sleep quality outcome measures were as follows: SOL, defined as time (minutes) between a detected commencement of a rest interval when lights were registered as ‘off’ and the sleep onset time; TST, defined as total time (hours) between sleep onset and offset; number of awakenings, defined as the total number of epoch blocks within the TST interval that were registered as awake; wake after sleep onset (WASO), calculated as the total duration of each awakening episode (minutes); and sleep efficiency (SE), defined as percentage of time spent in bed asleep relative to the total time spent in bed between getting into bed and getting up the following morning.

#### 2.4.3. Self-Reported Sleep

A participant questionnaire was developed to record self-reported details of the previous night’s sleep, akin to a single night of a traditional sleep diary, including SOL, TST, number of awakenings, and reasons for awakenings. The questionnaire was administered the next morning by a clinical physiotherapist. 

### 2.5. Statistical Analyses

SPSS (version 28.0, IBM, IL, USA) was used to test the level of agreement, within participants, between the Actiwatch and WSA for SOL, TST, number of awakenings, WASO, and SE via intraclass correlation coefficients (ICC) [18]. Absolute agreement with a two-way mixed effect model was used to determine average ICC. This approach was selected, as we assumed that the error from the devices would be predictable and fixed, and the error from the participants would be random. Agreement between the Actiwatch and participant-reported SOL, TST, and number of awakenings were also examined. ICC categories were established a priori to be poor (<0.50), moderate (0.50–0.75), good (0.75–0.90), and excellent (>0.90) [20]. Bland–Altman plots were generated to display the differences between the Actiwatch and WSA devices against the overall mean scores for the two devices, including the upper and lower 95% limits around the combined mean, consistent with previous studies examining the validity of sleep recording devices [14,18,21].

## 3. Results

Eighteen participants wore the Actiwatch for an entire night, with no participants removing the device from their wrist. Ten participants also used the WSA, but data for two participants could not be used due to device pairing issues, and no valid data were collected on the WSA from an additional two participants who slept in a chair instead of the hospital bed. Therefore, actigraph data were available for all 18 participants, and both actigraph and WSA data were available for a subsample of 6 participants. 

Participant characteristics are provided in Table 1. There was relative consistency in the characteristics within the entire cohort (n = 18) who used the actigraphy device as well as in the sub-sample who used both the actigraphy device and the WSA (n = 6). The characteristic with the greatest discrepancy was the ward type. Of the eight participants who had sleep monitored via both Actiwatch and WSA devices, just one of these participants was recruited from the specialized neurological ward, with the remaining seven recruited from the general ward (Table 1). 

Figure 1 depicts the sleep and wakening events for both the Actiwatch and WSA data for one participant, and similar comparisons are available in Appendix A for all other participants. Raw data for all participants who used both the Actiwatch and WSA are provided in Appendix A. 

ICC levels of agreement for each sleep parameter were poor between Actiwatch and participant-reported sleep quality and between the Actiwatch and WSA (Table 2). The Actiwatch underestimated sleep onset latency and over-estimated total sleep time and awakenings relative to both participant report and the WSA. Overall, there appeared to be poorer agreement between the Actiwatch and WSA than between the Actiwatch and self-report for sleep onset latency. The WSA also showed markedly lower sleep efficiency and wakenings after sleep onset relative to the Actiwatch. 

The Bland–Altman plots demonstrated a broad range in the difference between Actiwatch and WSA device data (Figure 2). Participants whose data fell outside of the 95% confidence interval around the combined mean between the Actiwatch and WSA typically had longer sleep onset latency, lower total sleep time, more awakenings, and lower sleep efficiency, highlighting that there was a potential bias towards poor agreement when people had worse sleep.

## 4. Discussion

This study found poor agreement for all sleep parameters between the low-cost WSA and widely used Actiwatch. This study also found poor agreement between participant-reported subjective sleep quality and the Actiwatch device parameters of sleep. Given that devices, such as actigraphy watches, measure different aspects of sleep compared with subjective reports of sleep quality, their poor level of agreement has been well-documented previously [22]. The results of the present study are clinically relevant, as they highlight a need for multiple modalities of effective sleep monitoring in the clinical setting. A low-cost alternative, such as the WSA, which could routinely monitor sleep quality in adults with stroke while undergoing rehabilitation, may inform and assist in guiding rehabilitation programs to optimize recovery; however, we found that this device also did not reliably agree with the well validated Actiwatch. 

It is worth noting that previous studies have also found that actigraphic recordings had poorer agreement with polysomnography than other commercially available devices, including an under-the-mattress device in healthy young adults [14]. Importantly, the study by Chinoy, Cuellar, Huwa, Jameson, Watson, Bessman, Hirsch, Cooper, Drummond and Markwald [14] tested the reliability of consumer wearable and “nearable” sleep tracking devices in conditions that could be considered to mimic the type and frequency of overnight disruptions that occur in an inpatient rehabilitation hospital setting. Chinoy, Cuellar, Huwa, Jameson, Watson, Bessman, Hirsch, Cooper, Drummond and Markwald [14] found that the “under-the-mattress” device that they tested overestimated total sleep time by approximately 14 min and sleep efficiency by 2.9%, and underestimated time spent awake after sleep onset by 15 min relative to PSG. Similarly, we found that the WSA device underestimated total sleep time by 1.2 h and overestimated time spent awake after sleep onset by 66 min. In order to enable appropriate selection and use of sleep monitoring devices, further large-scale studies comparing the accuracy and reliability of a range of consumer accessible low-cost devices, such as the WSA, against both actigraphy and polysomnography in the context of neurological rehabilitation are necessary. 

The discrepancies in sleep parameters between devices, and with self-report, may be due to a number of factors. Firstly, the small sample size may have been a contributing factor, as there was large variability in the data for each of the sleep parameters, and estimates could have been unduly influenced by outliers in the data. Future studies using larger samples of adults with stroke may reduce the observed data variability and develop further insight into whether the WSA and participant reports are comparable to the more expensive devices, which are held to be more accurate. Further, the discrepancies observed may also reflect the different activity thresholds of each device to calculate sleep parameters. Actiwatch activity thresholds are well-defined, readily available, and generally have reported high levels of agreement with polysomnography, particularly for detecting sleep time rather than wakenings, and is considered a ‘gold standard’ for sleep monitoring in settings where it is impractical to use polysomnography [18,21]. Conversely, activity thresholds are not readily available for many consumable devices, including the WSA. Obtaining transparency on the WSA activity thresholds will be key to determining whether it is comparable to established actigraphy devices in future research. 

The actigraphy methodology used in this study differed from previous studies monitoring sleep quality in adults with stroke. Firstly, as this was a study investigating the feasibility and usability of the Actiwatch and WSA devices prior to further larger-scale studies, sleep was monitored over one night only. This is not consistent with recent recommendations that sleep monitoring via actigraphy be conducted over a 7–14-day period to account for night-to-night variability in sleep parameters within an individual [23,24]. These recommendations, however, are more relevant to studies investigating clinical understandings of sleep quality. Given that the current validation study focused on the agreement between these two devices, recommendations to monitor sleep for more than one night may not apply in this context. 

While the activity thresholds and sleep parameters were comparable, the limb wearing the Actiwatch differed from previous studies. The Actiwatch was placed on the hemiparetic limb for safety reasons, which differed from previous recent studies investigating similar cohorts of adults with stroke [2,8,9,25]. This may have influenced the accuracy of sleep measurements. However, we found that SOL was longer than in previous studies [2,25], TST was comparable to one study [8] yet longer than others [2,25], while WASO was comparable to that found in similar studies [9], shorter than one [25], and longer than another [2]. Potential lower levels of movement in the hemiparetic limb overnight, awakenings in the hospital environment, and WASO may have been underestimated, as wakeful periods overnight may not have resulted in movements of the affected limb. Further, SOL and TST may be overestimated as participants may waken prior to sufficient movements in the hemiparetic limb can trigger the Actiwatch to register that they are awake. However, all participants were able to move independently in and out of bed, minimizing the risk of this occurring. Rather, it appeared that the participants who had poorer agreement between devices had worse sleep (i.e., longer sleep onset latency, lower total sleep time, more awakenings, and lower sleep efficiency), similar to previous studies examining the agreement between consumer sleep monitoring devices with polysomnography [14]. Moreover, it is likely that the discrepancies between devices were due to the small sample size, the inherent variability of one night of sleep monitoring, and the potential differing activity thresholds and sensitivity between devices, which may be more pronounced in people with hemiparesis. 

### 4.1. Device Usabiltiy

The non-wearable WSA may offer a low-cost alternative to the Actiwatch device, measuring sleep from cardiac activity, breathing patterns, and snoring in addition to body movements. However, a number of usability issues arose that impacted its application. The WSA required continuous power supply via a plug-in power source at the hospital bedside. Additionally, the device relied upon application-based data storage, so Bluetooth pairing capabilities and reliable internet network connectivity was required on the hospital wards. Given the often-unreliable network connectivity in the hospital environment and variability in proximity of bedside power sources, these requirements presented challenges to data collection, and contributed to the low sample size. Moreover, some patients clearly prefer to sleep in a recliner chair rather than the hospital bed overnight where the WSA device cannot monitor their sleep. 

### 4.2. Limitations

As discussed above, there are a number of limitations to the study that may influence the interpretation our findings. The small sample size, single night of sleep monitoring, and actigraphy sleep monitoring on the hemiparetic limb may have contributed to the reported poor level of agreement between the WSA and Actiwatch. However, despite our small sample size, the 95% confidence intervals for the Bland–Altman tests were not dissimilar to those published by Chinoy, Cuellar, Huwa, Jameson, Watson, Bessman, Hirsch, Cooper, Drummond and Markwald [14] for a similar “under-the-mattress” device compared with polysomnography in 19 healthy young adults. While the WSA measures sleep from physiological and behavioral signals in addition to bodily movements, there was very large variability in the WSA estimates relative to the Actiwatch, which determines sleep metrics from limb movements only. These differences may be even more pertinent for people who have had a stroke with hemiparesis. Following on from the methodology developed in this current feasibility study, future studies addressing these limitations by increasing the sample size and wearing the Actiwatch on the non-hemiparetic limb may assist in obtaining new insights into the validity and utility of the WSA as a low-cost alternative to actigraphy for effective sleep monitoring in the hospital environment.

## 5. Conclusions

The findings from the present study suggest potential issues with the usability and accuracy of the WSA for monitoring sleep quality in adults who have been admitted to hospital for inpatient rehabilitation following a stroke. Future studies in larger samples and across multiple nights are needed to further investigate whether under-the-mattress technology can consistently and accurately monitor sleep parameters in adults with stroke. Moreover, developers of consumer devices for monitoring sleep should enable researchers to access their raw data so that the tools can be independently validated for reliable measurement of sleep in research settings and in unique clinical populations. Combining the use of wearable or under-the-mattress devices with subjective reports of sleep quality by adults using standardized outcome measures, together with review of clinical notes on patient sleep quality, may enable low-cost assessment of sleep in an inpatient rehabilitation setting. This may ultimately assist in understanding the nature and impact of sleep disturbances following stroke. Such studies may also enable the development of strategies to reduce the impact of the hospital environment on sleep quality, helping patients to have optimal engagement in rehabilitation programs to facilitate their recovery from stroke. 

## Figures and Tables

**Figure 1 ijerph-20-03984-f001:**
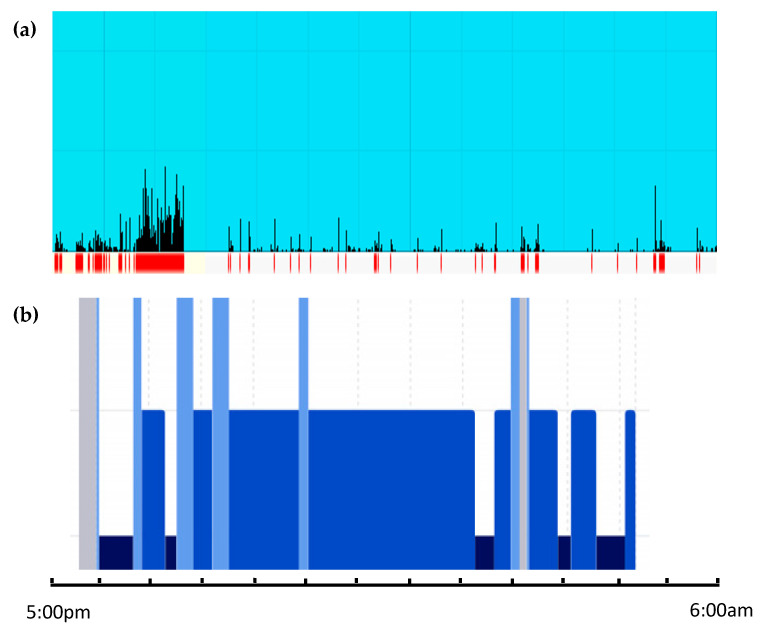
Sleep activity for one participant. (**a**) Philips Actiwatch: Blue shaded region represents time asleep; black bar graph indicates movement and activity overnight; and red lines indicate awake periods. (**b**) WSA: Blue and black regions represent sleep and deep sleep periods respectively; grey regions represent awake periods.

**Figure 2 ijerph-20-03984-f002:**
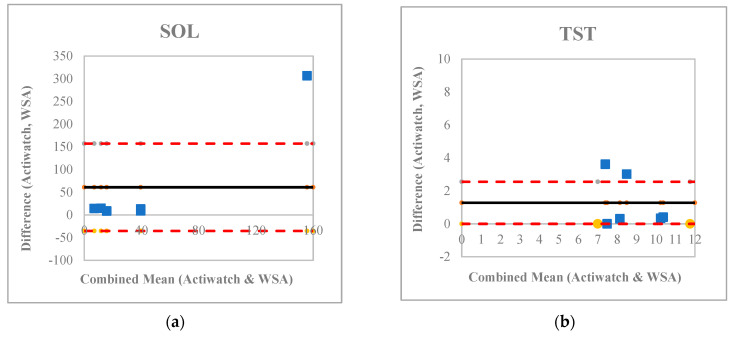
Bland-Altman plots comparing Actiwatch and WSA devices for (**a**) Sleep onset latency, (**b**) Total sleep time, (**c**) Number of awakenings, (**d**) Wake after sleep onset, and (**e**) Sleep efficiency. Blue dots indicate differences between the two devices against the combined mean for each participant. Solid black lines represent overall mean for the two devices. Dashed red lines represent the mean difference upper and lower 95% CI for the overall combined mean.

**Table 1 ijerph-20-03984-t001:** Descriptive statistics and participant characteristics.

Characteristic	Total Sample (n = 18)	Matched Sub-Sample (n = 6)
Age (yrs), mean ± SD	72.2 ± 10.0	69.2 ± 14.5
Male sex, n (%)	12 (66.7)	3 (50)
Ischemic stroke, n (%)	14 (77.8)	5 (83.3)
Stroke location, n (%)		
Brain Stem	7 (38.9)	3 (50)
Basal Ganglia	3 (16.7)	1 (16.7)
Temporal/Parietal	3 (16.7)	0 (0)
Middle Cerebral Artery	2 (11.1)	1 (16.7)
Internal Capsule	1 (5.6)	0 (0)
Cerebellum	1 (5.6)	0 (0)
Multi-territorial	1 (5.6)	1 (16.7)
Neurological rehabilitation ward, n (%)	8 (44.4)	1 (16.7)
Days since stroke	31 ± 22	31.5 ± 25.5
Days since rehabilitation admission, n (%)		
0–1 weeks	6 (33.3)	3 (50)
1–2 weeks	4 (22.2)	1 (16.7)
2–4 weeks	4 (22.2)	0 (0)
1–2 months	1 (11.1)	1 (16.7)
2–3 months	3 (33.3)	1 (16.7)

**Table 2 ijerph-20-03984-t002:** Measurement of sleep quality.

	Total Sample (n = 18)			Matched Sub-Sample (n = 6)		
Outcome Measure	Actiwatch	Participant-Reported		Actiwatch	WSA	
	Median (IQR)	Median (IQR)	ICC (95% CI)	Median (IQR)	Median (IQR)	ICC (95% CI)
SOL (min)	19.3 (38.6)	30 (20)	−0.28 (−1.26, 0.76)	15.9 ± 18.2	27.5 (24)	−0.17 (−5.38, 0.72)
TST (h)	8.6 (2.5)	7 (2.5)	0.31 (−0.743, 0.73)	9.3 ± 1.3	7.9 (2.5)	0.52 (−0.81, 0.92)
Awakenings (count)	23 (9.8)	3 (1.75)	0.02 (−0.719, 0.34)	19.3 ± 8.75	3.5 (5.3)	0.21 (−0.24, 0.78)
WASO (min)	24.8 (18.8)			20.9 ± 10.8	62. (68.3)	0.23 (−1.46, 0.87)
SE (%)	88.3 (17.3)			89.1 ± 7.4	83 (22.8)	−0.07 (−3.08, 0.83)

* level of agreement *p* < 0.05, which was not met in any of the analyses. Abbreviations: CI = Confidence interval; Hrs = Hours; ICC = Intraclass Correlation Coefficient; IQR = Interquartile Range; Mins = Minutes; SE = Sleep Efficiency; SOL = Sleep Onset Latency; TST = Total Sleep Time; WASO = Wake after sleep onset; and WSA = Withings sleep analyzer69.

## Data Availability

Not applicable.

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
