# Peer review of "Measuring Sleep Quality in the Hospital Environment with Wearable and Non-Wearable Devices in Adults with Stroke Undergoing Inpatient Rehabilitation"

_ijerph, 2023, doi:10.3390/ijerph20053984_

Round 1
Reviewer 1 Report
The paper is very timely and attempts to compare well established wearables to new technology and cheaper options that potentially can provide easily accessible solutions for clinical settings. However, the high variability between the groups in numbers and population characteristics makes it rather difficult to draw meaningful conclusions. Perhaps the paper would benefited if the authors included some closely explore case studies drawing from the participants at hand.
Introduction
1. The authors state that “Wearable actigraphy devices are non-invasive technologies that are able to monitor and accurately quantify sleep quality parameters.” Sleep quality is a rather subjective variable and not a component that can be quantify or measured using actigraphy, which is in essence an objective tool.
2. Rather than opportunity, why the researchers chose to compare WSA to Actigraphy? What is the unique features of the tool that would be suitable for sleep research in this cohort?
Methods
3. The researchers collected subjective account of the previous’ night sleep, which in essence is a sleep diary. However, this is reported as “Sleep parameters, self report” and not sleep diary. Is there a particular reasoning behind this distinction?
Results
4. The in-between group analyses is rather ambitious given the high variability between the groups in power and characteristic similarity. It is also, not clear why the researchers decided to depict the agreement difference between the methods rather than a Tukey plot.
5. More content to the figure 2 should be helpful for the reader, perhaps depicting the comparison in a case study format.
Author Response
The authors state that “Wearable actigraphy devices are non-invasive technologies that are able to monitor and accurately quantify Sleep quality parameters.” Sleep quality is a rather subjective variable and not a component that can be quantify or measured using actigraphy, which is in essence an objective tool.
RESPONSE: This is an excellent point. We have revised the sentence, including removal of the term “quality".
Rather than opportunity, why the researchers chose to compare WSA to Actigraphy? What is the unique features of the tool that would be suitable for sleep research in this cohort?
RESPONSE: We have incorporated additional explanation of why a device that is placed under the mattress may offer benefits in accurate recordings of sleep in people who have had a stroke (Introduction, lines 62-67).
The researchers collected subjective account of the previous’ night sleep, which in essence is a sleep diary. However, this is reported as “Sleep parameters, self report” and not sleep diary. Is there a particular reasoning behind this distinction?
RESPONSE: We have rephrased the text in section 2.4.3, highlighting that the method was similar to completion of a sleep diary for a single night.
The in-between group analyses is rather ambitious given the high variability between the groups in power and characteristic similarity. It is also, not clear why the researchers decided to depict the agreement difference between the methods rather than a Tukey plot.
RESPONSE: The study examined within subject agreement, and did not undertaken any between group analyses. To make sure that this is clear to readers we have made updates to the
- Introduction (line 78-79, “to explore the level of agreement between a validated actigraphy device and the WSA within a sample of adults”).
- Study design section (line 84, “Cross-sectional within subjects cohort study”
- Statistical analysis section (line 151-152, “test the level of agreement, within participants, between the Actiwatch and WSA”)
- Results, cohort overview (line 174-176, “characteristics between the entire cohort (n=18) that used the actigraphy device, and the sub-sample who used both the actigraphy device and the WSA (n=6)”
More content to the figure 2 should be helpful for the reader, perhaps depicting the comparison in a case study format.
RESPONSE: The case data for every participant are already provided in Figure 1 and Supplementary Figure 1-5. These figures depcit every sleep period, awakening, and movement/activity event as recorded by each device. The purpose of Figure 2 is to show the level of agreement/difference for each of the key sleep parameters between devices for all participants. We have expanded the interpretation of Figure 2 in text to help highlight how and where individual partipants fall outside of the 95% confidence intervals, depicting poor agreement.
Reviewer 2 Report
Review
Introduction
The introduction provides a good background of the topic, highlighting the role of good sleep for in-hospital stroke patients especially for rehabilitation candidates. Sleep duration and sleep quality influence the functional outcome after a stroke. Collecting data about sleep can be difficult and expensive in hospital wards but nowadays there are many devices that can be used with ease. The authors make a comparison between a common actigraphy device (Actiwatch) and the Withings Sleep Analyzer (WSA). I suggest emphasizing that the WSA is primarily tested in the diagnosis of obstructive sleep apnea syndrome and that current literature is scarce about this device. The aim of the study is clear.
Material and methods
The cross-sectional design of the study is coherent with the aim. In the study participants sections I suggest highlighting the reason for recruiting patients from general and neurological wards (are there differences in sleep settings, lights, time schedule?). In the sub-section 2.3.1 on the line 71, I suggest replacing hemiplegic with hemiparetic because it is stated in the previous section that patients were able to move and autonomously roll in bed.
In subsection 2.4.1 would be interesting to show if patients were on sedating drugs or other kinds of
medications that could interfere with sleep.
In subsection 2.6 statistical analyses would be useful to provide more details about the kind of intra class correlation that you used. Please look at the following article (Koo TK, Li MY. A Guideline of Selecting and Reporting Intraclass Correlation Coefficients for Reliability Research. J Chiropr Med. 2016 Jun;15(2):155-63. doi: 10.1016/j.jcm.2016.02.012. Epub 2016 Mar 31. Erratum in: J Chiropr Med. 2017 Dec;16(4):346. PMID:27330520; PMCID: PMC4913118).
Results
You should explain better the sentence in the line 123 -125.
Figures and tables are easy to understand. In the supplementary materials there are only 5 figures representing the six sub-sampled patients, is there a missing figure or is that in the text? In table S1 the participant reported data for patients n.6 are missing.
Discussion
The differences and poor agreements between the two devices are well explained.
Could be useful to your thesis to add something from this recent study investigating differences between various sleep trackers, wearable and non, versus standard PSG. (Chinoy ED, Cuellar JA, Huwa KE, Jameson JT, Watson CH, Bessman SC, Hirsch DA, Cooper AD, Drummond SPA, Markwald RR. Performance of seven consumer sleep-tracking devices compared with polysomnography. Sleep. 2021 May 14;44(5):zsaa291. doi:10.1093/sleep/zsaa291. PMID: 33378539; PMCID: PMC8120339.)
In the line 172 the term differing is a refuse?
The sentences between line 199 and 202 are not very clear, could you explain it better?
About the subsection 4.2, I suggest adding a small sentence about the intrinsic differences between the two devices: e.g. the WSA is an under mattress device which is less influenced by the ability to move the limb with respect to the Actiwatch.
Conclusions
Please explain better and more clearly the sentence on the line 247 - 250.
The remaining part of the section is well written and easy to understand. Nonetheless the WSA lacks the possibility of accessing the raw data collected: improving this aspect could be useful in clinical and research settings.
Author Response
I suggest emphasizing that the WSA is primarily tested in the diagnosis of obstructive sleep apnea syndrome and that current literature is scarce about this device.
RESPONSE: We have expanded the description of the use of the WSA to date for diagnosis of sleep apnea (lines 69-75).
In the study participants sections I suggest highlighting the reason for recruiting patients from general and neurological wards (are there differences in sleep settings, lights, time schedule?).
RESPONSE: We have expanded the description of the two rehabilitation wards. This now details the rationale for recruiting across two rehabilitation wards, and highlights the key differences between the wards with respect to monitoring sleep (lines 89-95).
In the sub-section 2.3.1 on the line 71, I suggest replacing hemiplegic with hemiparetic because it is stated in the previous section that patients were able to move and autonomously roll in bed.
RESPONSE: This change has been made throughout the manuscript, as suggested.
In subsection 2.4.1 would be interesting to show if patients were on sedating drugs or other kinds of medications that could interfere with sleep.
RESPONSE: We can confirm that no patients were on sedating medications. This has been added to the methods section (line 132).
In subsection 2.6 statistical analyses would be useful to provide more details about the kind of intra class correlation that you used. Please look at the following article (Koo TK, Li MY. A Guideline of Selecting and Reporting Intraclass Correlation Coefficients for Reliability Research. J Chiropr Med. 2016 Jun;15(2):155-63. doi:10.1016/j.jcm.2016.02.012. Epub 2016 Mar 31. Erratum in: JChiropr Med. 2017 Dec;16(4):346. PMID:27330520; PMCID:PMC4913118).
RESPONSE: We apologise and have now added in the ICC type (two-way mixed effect) on line 153-154. Please note that we have also updated the data reported in Table 2 as we realised that reporting the mean and standard deviation was not appropriate with such high variability in the dataset. Instead we now report the Median and IQR values.
You should explain better the sentence in the line 123 -125.
RESPONSE: We have completely reworded this sentence to make sure that it is clear to readers.
In the supplementary materials there are only 5 figures representing the six sub-sampled patients, is there a missing figure or is that in the text? In table S1 the participant reported data for patients n.6 are missing.
RESPONSE: We have revised the text so that it is clear that the graphical comparison of the Actiwatch and WSA data are provided for all six participants, one in the main text of the paper and five in supplementary materials (see lines 183-187).
Could be useful to your thesis to add something from this recent study investigating differences between various sleep trackers, wearable and non, versus standard PSG. (Chinoy ED, Cuellar JA, Huwa KE, Jameson JT, Watson CH, Bessman SC, Hirsch DA, Cooper AD, Drummond SPA, Markwald RR. Performance of seven consumer sleep-tracking devices compared with polysomnography. Sleep. 2021 May 14;44(5):zsaa291. doi:10.1093/sleep/zsaa291.PMID: 33378539; PMCID: PMC8120339.)
RESPONSE: We have expanded our discussion of the literature, including the study by Chinoy et al. in the discussion.
In the line 172 the term differing is a refuse?
RESPONSE: It appears that the wrong line number has been cited by the reviewer; however, we suspect the relevant text was line 4 of the discussion. We have updated this sentence so that it more clearly describes the differences between objective measures of sleep and subjective ratings of sleep quality.
The sentences between line 199 and 202 are not very clear, couldyou explain it better?
RESPONSE: As above, it appears that the wrong lines have been cited. However, we have checked the whole discussion section for clarity and made minor revisions throughout to improve clarity.
About the subsection 4.2, I suggest adding a small sentence aboutthe intrinsic differences between the two devices: e.g. the WSA isan under mattress device which is less influenced by the ability tomove the limb with respect to the Actiwatch.
RESPONSE: The limitations section has been updated as suggested, with the inclusion of the following sentences (line 370-374):
“While the WSA measures sleep from physiological and behavioral signals in addition to bodily movements, there was very large variability in the WSA estimates relative to the Actiwatch, which determines sleep metrics from limb movements only. These differences may be even more pertinent for people who have had a stroke with hemiparesis.”
Conclusions: Please explain better and more clearly the sentence on the line 247- 250.
RESPONSE: As above, the line numbers are not correct; however, we have revised the conclusions section to ensure that the text is clear.
The remaining part of the section is well written and easy to understand. Nonetheless the WSA lacks the possibility of accessing the raw data collected: improving this aspect could be useful in clinical and research settings.
RESPONSE: We have added a sentence to the discussion highlighting this point (line 306-308):
“Moreover, developers of consumer devices for monitoring sleep should enable researchers to access the raw data so that the tools can be independently validated for reliable meas-urement of sleep in research settings and in unique clinical populations.”
Round 2
Reviewer 1 Report
I'm pleased to see the changes and revisions by the authors to the recommended suggestions. The paper will be a strong contribution to the field.